# ON LEARNING HETEROSCEDASTIC NOISE MODELS WITHIN DIFFERENTIABLE BAYES FILTERS

## ABSTRACT

In many robotic applications, it is crucial to maintain a belief about the state of a system, like the location of a robot or the pose of an object. These state estimates serve as input for planning and decision making and provide feedback during task execution. Recursive Bayesian Filtering algorithms address the state estimation problem, but they require a model of the process dynamics and the sensory observations as well as noise estimates that quantify the accuracy of these models. Recently, multiple works have demonstrated that the process and sensor models can be learned by end-to-end training through differentiable versions of Recursive Filtering methods. However, even if the predictive models are known, finding suitable noise models remains challenging. Therefore, many practical applications rely on very simplistic noise models. Our hypothesis is that end-to-end training through differentiable Bayesian Filters enables us to learn more complex *heteroscedastic* noise models for the system dynamics. We evaluate learning such models with different types of filtering algorithms and on two different robotic tasks. Our experiments show that especially for sampling-based filters like the Particle Filter, learning heteroscedastic noise models can drastically improve the tracking performance in comparison to using constant noise models.

## 1 INTRODUCTION

For many real-world systems that we would like to control, we cannot directly observe the current state directly. However, in order to stabilize a system at a goal state or make it track a trajectory, we need to have access to state feedback. An observer provides an estimate of the current system state from sensor measurements. *Recursive Bayesian Filtering* is a probabilistic approach towards estimating a belief about the current state. The method relies on a process model that predicts how the system behaves over time and an observation model that generates the expected observations given the predicted state. While the approach itself is general and makes few assumptions, the challenge is to formulate the process and observation models and to estimate the noise in these models. Process and observation noise quantify how certain the filter is about either the prediction or the observations. This information is used to determine how much the predicted state is updated based on the observation.

Deep neural networks are well suited for tasks that require finding patterns or extracting information from raw, high-dimensional input signals and compressing them into a more compact representation. They have therefore become the method of choice especially in perception problems. For many robotics tasks like modeling dynamics, planning or tracking however, it has been shown that combining prior knowledge in the form of analytical models and/or algorithmic structure with trainable network components leads to better performance and generalizability than trying to learn the complete tasks from scratch (Kloss et al., 2017; Karkus et al., 2017; Jonschkowski et al., 2018; Tamar et al., 2016; Okada et al., 2017; Jonschkowski & Brock, 2016; Haarnoja et al., 2016; Karkus et al., 2018).

Specifically, (Jonschkowski & Brock, 2016; Haarnoja et al., 2016; Jonschkowski et al., 2018; Karkus et al., 2018) have presented differentiable Bayesian Filtering algorithms. The authors focus on learning the observation and dynamics models end-to-end through the filters and demonstrate that the recursive filtering structure improves prediction results over using recurrent neural networks that were trained for the same task.

In many robotic applications, it is possible to formulate the process and observation model based on first-order principles. However, finding appropriate values for the process and observation noise is often difficult and despite of much research on identification methods (e.g. (Bavdekar et al., 2011; Valappil & Georgakis, 2000)) they are often tuned manually. To reduce the tedious tuning effort, the noise models are typically assumed to be a Gaussian with zero mean and constant covariance. Many real systems can however be better modeled with *heteroscedastic* noise models, where the level of uncertainty depends on the state of the system and/or possible control inputs. Taking heterostochasticity into account has been demonstrated to improve filtering performance in many robotic tasks (Bauza & Rodriguez, 2017; Kersting et al., 2007).

In this work, we propose a method to learn heteroscedastic noise models from data by optimizing the prediction likelihood end-to-end through differentiable Bayesian Filters. In addition to differentiable Extended Kalman Filters and Particle Filters, which have been proposed in related work, we also propose two different versions of the Unscented Kalman Filter.

In our experiments we focus on learning the noise models and therefore assume that observation and process models are known or at least pretrained. We evaluate the performance of the different filters and noise models on two different real-world robotic problems: (i) Visual Odometry for an driving car (Haarnoja et al., 2016; Jonschkowski et al., 2018; Geiger et al., 2012) which has simple smooth dynamics and a low-dimensional state, and (ii) Visual tracking of an object that is pushed by a robot (Yu et al., 2016; Kloss et al., 2017). Planar pushing has challenging, discontinuous dynamics and was shown to have a heteroscedastic noise distribution (Bauza & Rodriguez, 2017). Furthermore, the dimensionality of the state is double of the Visual Odometry task.

Our experiments show that using heteroscedastic process noise models drastically improves the tracking performance of the Particle Filter and Unscented Filter variants and facilitated learning as compared to learning a constant process noise model. While learning the noise models can be beneficial for all filters, the tracking performance of the EKF turned out to be least sensitive to the noise models. In comparison to the process noise, learning the observation noise did not improve the results much for the two tasks we evaluated.

## 2 Background: Bayesian Filtering

*Filtering* refers to the problem of estimating the state $\mathbf{x}$ of a stochastic system at time step $t$ given an initial believe $\mathbf{x}_0$, a sequence of observations $\mathbf{z}_t$ and control inputs $\mathbf{u}_t$. The aim is to compute $p(\mathbf{x}_t|\mathbf{x}_{0...t-1}, \mathbf{u}_{0...t}, \mathbf{z}_{0...t})$. To do so, we describe the system with a state space representation, that consists of two equations: The *process model* $f$ describes how the state changes over time and the *observation model* $h$ generates observations given the current state:

$$\mathbf{x}_t = f(\mathbf{x}_{t-1}, \mathbf{u}_{t-1}, \mathbf{q}_t) \qquad\qquad \mathbf{z}_t = h(\mathbf{x}_t, \mathbf{r}_t)$$

The random variables $\mathbf{q}$ and $\mathbf{r}$ are the process and observation noise and represent the stochasticity of the system. This model makes the Markov assumption, i.e. the current state only depends on the previous state, and the observation only depends on the current state. This assumption makes it possible to compute $p(\mathbf{x}_t|\mathbf{x}_{0...t-1}, \mathbf{u}_{0...t}, \mathbf{z}_{0...t})$ recursively from $p(\mathbf{x}_{t-1}|\mathbf{x}_{0...t-2}, \mathbf{u}_{0...t-1}, \mathbf{z}_{0...t-1})$. In the following, we review the most common filtering algorithms. For more details, we refer to Thrun et al. (2005).

### 2.1 Kalman Filter

The Kalman Filter (Kalman, 1960) is a closed-form solution to the filtering problem for systems with linear process and observation model and Gaussian additive noise.

$$\mathbf{x}_t = f(\mathbf{x}_{t-1}, \mathbf{u}_{t-1}, \mathbf{q}_t) = \mathbf{A}\mathbf{x}_{t-1} + \mathbf{B}\mathbf{u}_t + \mathbf{q}_t \qquad \mathbf{z}_t = h(\mathbf{x}_t, \mathbf{r}_t) = \mathbf{H}\mathbf{x}_t + \mathbf{r}_t$$

Given these assumptions and a Gaussian initial belief, the belief can be represented by the mean $\mu$ and covariance matrix $\Sigma$ over the estimate. At each timestep, the filter predicts $\hat{\mu}$ and $\hat{\Sigma}$ given the process model. The innovation $\mathbf{i}_t$ is the difference between the predicted and actual observation and is used to correct the prediction. The Kalman Gain $K$ trades-off the process noise $Q$ and the observation noise $R$ to determine the magnitude of the update.

Prediction Step: Update Step:

$$\hat{\boldsymbol{\mu}}_t = \mathbf{A}\boldsymbol{\mu}_{t-1} + \mathbf{B}\mathbf{u}_t \quad (1) \qquad \mathbf{S}_t = \mathbf{H}\hat{\boldsymbol{\Sigma}}_t\mathbf{H}^T + \mathbf{R}_t \quad (3) \qquad \mathbf{i}_t = \mathbf{z}_t - \mathbf{H}\hat{\boldsymbol{\mu}}_t \quad (5)$$

$$\hat{\boldsymbol{\Sigma}}_t = \mathbf{A}\boldsymbol{\Sigma}_{t-1}\mathbf{A}^T + \mathbf{Q}_t \quad (2) \qquad \mathbf{K}_t = \hat{\boldsymbol{\Sigma}}_t\mathbf{H}^T\mathbf{S}_t^{-1} \quad (4) \qquad \boldsymbol{\mu}_t = \hat{\boldsymbol{\mu}}_t + \mathbf{K}_t\mathbf{i}_t \quad (6)$$

$$\boldsymbol{\Sigma}_t = (\mathbf{I}_n - \mathbf{K}_t\mathbf{H})\hat{\boldsymbol{\Sigma}}_t \quad (7)$$

## 2.2 EXTENDED KALMAN FILTER (EKF)

The EKF (Sorenson, 1985) extends the Kalman Filter to systems with non-linear process and observation models. It uses the non-linear models for predicting $\hat{\mu}$ and the corresponding observations $\hat{\mathbf{z}}$ (Equations 1, 5). For computing the prediction and update of $\Sigma$ and $\mathbf{K}$, these models are linearized around the current mean of the state and the Jacobians $\mathbf{F}_{|\mu_t}$ and $\mathbf{H}_{|\mu_t}$ replace $\mathbf{A}$ and $\mathbf{H}$ in Equations 2 - 4 and 7. This first-order approximation can be problematic for systems with strong non-linearity, as it does not take the uncertainty about the mean into account (Van Der Merwe, 2004).

## 2.3 UNSCENTED KALMAN FILTER (UKF)

The UKF (Simon J. Julier, 1997; Van Der Merwe, 2004) was proposed to address the aforementioned problem of the EKF. Its core idea is to represent a Gaussian random variable by a set of specifically chosen points in state space, the so called *sigma points* $\mathcal{X}$. If this random variable undergoes a nonlinear transformation, we can calculate its new statistics from the transformed sigma points. This method is called the *Unscented Transform* (Simon J. Julier, 1997). For example, in the prediction step of the UKF, the non-linear transform is the process model (Equation 10) and the new mean and covariance are computed in Equations 11 and 12

$$\mathcal{X}^0 = \boldsymbol{\mu} \qquad\qquad \mathcal{X}^i = \boldsymbol{\mu} \pm (\sqrt{(n+\lambda)\boldsymbol{\Sigma}})_i \qquad \forall i \in \{1...n\} \quad (8)$$

$$w^0 = \frac{\lambda}{\lambda+n} \qquad\qquad w^i = \frac{0.5}{\lambda+n} \qquad\qquad \forall i \in \{1...2n\} \quad (9)$$

$$\hat{\mathcal{X}}_t = f(\mathcal{X}_{t-1}, \mathbf{u}_t) \quad (10) \qquad \hat{\boldsymbol{\Sigma}}_t = \sum_i w^i (\hat{\mathcal{X}}_t^i - \hat{\boldsymbol{\mu}}_t)(\hat{\mathcal{X}}_t^i - \hat{\boldsymbol{\mu}}_t)^T + \mathbf{Q}_t \quad (12)$$

$$\hat{\boldsymbol{\mu}}_t = \sum_i w^i \hat{\mathcal{X}}_t^i \quad (11)$$

By applying the non-linear prediction step separately to each sigma point and then fitting a new Gaussian to the transformed points (Equations 11, 12), the UKF conveys the non- linear transformation of the covariance more faithfully than the EKF and is thus better suited for strongly non-linear problems (Thrun et al., 2005).

The parameter $\lambda$ controls the spread of the sigma points and how strongly the original mean $\mathcal{X}^0$ is weighted in comparison to the other sigma points. In practice, we found $\lambda$ difficult to tune since placing the sigma points too far from the mean increases prediction uncertainty and can even destabilize the filter. Simon J. Julier (1997) suggested to chose $\lambda$ such that $\lambda + n = 3$. This however results in negative values of $\lambda$ if $n > 3$, for which the estimated covariance matrix is not guaranteed to be positive semidefinite anymore (Simon J. Julier, 1997). In addition, $\mathcal{X}^0$, which represents the original mean, is weighted negatively in this case, which seems counterintuitive and can cause divergence of the estimated mean.

## 2.4 MONTE CARLO UNSCENTED KALMAN FILTER (MCUKF)

The UKF represents the belief over the state with as few sigma points as possible. However, as described above, finding the correct scaling parameter $\lambda$ can be difficult, especially if the state is high dimensional. Instead of relying on the unscented transform to calculate the mean and covariance of

the state at the next timestep, we can also resort to Monte Carlo methods, as proposed by Wüthrich et al. (2016). In practice, this means that we replace the carefully constructed sigma points and their weights in Equations 8, 9 with samples from the current estimated state distribution, which all have uniform weights. The rest of the UKF algorithm stays the same, but more samples are necessary to represent the distribution accurately.

## 2.5 PARTICLE FILTER (PF)

In contrast to the different variants of the Kalman Filter explained before, the Particle Filter (Gordon et al., 1993) does not assume a parametric representation of the state distribution. Instead, it represents the state with a set of *particles*. The particle-based representation allows the filter to track multiple hypotheses about the state at the same time and makes it a popular choice for tasks like localization or visual object tracking (Thrun et al., 2005).

An initial set of particles $\mathcal{X}_0$ is drawn from some prior belief and initialized with uniform weights. At each recursion step, new particles are generated by sampling process noise and applying the process model to the previous particles:

$$\mathcal{X}_t = f(\mathcal{X}_{t-1}, \mathbf{u}_t, \mathbf{q}_t) \tag{13}$$

For each observation $\mathbf{z}_t$, we then evaluate the likelihood $p(\mathbf{z}_t|\mathbf{x}_t^i)$ of a particle $\mathbf{x}_t^i$ having generated this observation. Based on this, the weight $w^i$ of each particle is updated: $\mathbf{x}^i$: $w_t^i = w_{t-1}^i p(\mathbf{z}_t|\mathbf{x}_t^i)$.

A common problem of this filter is particle deprivation: Over time, many particles will receive a very low likelihood $p(\mathbf{z}_t|\mathbf{x}_t^i)$, and the state would be represented by too few particles with high weights. To prevent this, the particle filter algorithm uses resampling, where a new set of particles with uniform weights is drawn (with replacement) from the old set, according to the weights. This step focuses the particle set on regions of high likelihood and is usually applied after each timestep.

## 3 RELATED WORK

### DIFFERENTIABLE FILTERING

Haarnoja et al. (2016) proposed the BackpropKF, a differentiable implementation of the Kalman Filter. While the observation and process model were assumed to be known, the differentiable implementation enabled the authors to train a neural network through the filter to preprocess the input images. This network can be viewed as a trainable part of the sensor which extracts the relevant information from the high-dimensional raw input data and also predicts the observation noise $\mathbf{R}$ dependent on the images. This heteroscedastic observation noise model was shown to be useful in situations where the desired information could not be extracted from the image, e.g. when a tracked object is occluded. BackpropKF outperformed an LSTM model that was trained to perform the same tasks due to the prior knowledge encoded in the Filtering algorithm and the given models.

Jonschkowski & Brock (2016) presented a differentiable Histogram Filter for discrete localization tasks in one or two dimensions. For this low-dimensional problem , both, the observation and the process model, were trained through the filter in a supervised or unsupervised manner. Experiments showed that optimizing the models end-to-end through the filter improved results on the metric that was optimized during training (MSE or localization accuracy) in comparison to filtering with models that were trained in isolation.

Jonschkowski et al. (2018); Karkus et al. (2018) proposed differentiable Particle Filters for localization and tracking of a mobile robot. In each work, a neural network was trained to predict the likelihood $p(\mathbf{z}_t|\mathbf{x}_t^i)$ of each particle given an image and a map of the environment. While (Karkus et al., 2018) used a given process model, Jonschkowski et al. (2018) learned the process model and the distribution from which the process noise is sampled. They however did not evaluate their method when only the process model or only the noise was learned and it is thus not clear how each of these two components individually affected the overall error rate of the filter. Karkus et al. (2018) additionally introduced soft resampling and thereby enabled backpropagation through more than one time step.

Related work demonstrated that (i) integrating algorithmic structure with learning leads to better results than training unconstrained networks and that (ii) it is possible and beneficial to train the

components of the filters end-to-end instead of in isolation. Each work focused on creating a differentiable version of a particular filtering algorithm.

In this work, we propose to learn heteroscedastic noise models and analyze the benefit of these models within different filtering algorithms and for two different applications.

All previous work described above was evaluated on tracking and visual odometry problems with low-dimensional states and observations (at most five dimensions) and smooth, although non-linear dynamics models. In contrast to this, we additionally evaluate the methods on a planar pushing task which has challenging non-linear and discontinuous dynamics due to physical contact and a 10-dimensional state space.

### VARIATIONAL INFERENCE

Variational methods provide an alternative way of learning the parameters of a probabilistic generative model and performing inference on its latent states. The main idea of variational inference is to approximate the intractable posterior distribution $p_\theta(\mathbf{x}|\mathbf{z})$ by an approximate distribution $q_\phi(\mathbf{x}|\mathbf{z})$ that is easier to compute, e.g. because it factorizes over variables. The parameters $\theta$ of the true generative distribution $p$ (i.e. the model parameters) can be optimized jointly with the parameters $\phi$ of the approximate distribution $q$ by maximizing the evidence lower bound (ELBO).

Fraccaro et al. (2017) combine a locally linear gaussian state space model (LGSSM) with a variational autoencoder (Kingma & Welling, 2013; Rezende et al., 2014) that learns to encode high-dimensional sensory input data into a low-dimensional latent representation in an unsupervised way. This latent encoding is used as observations to the LGSSM. In addition, an LSTM network predicts the parameters of the process model from the history of latent encodings. Watter et al. (2015) follow a similar approach. Their method predicts a latent representation as well as the parameters of a a locally linear gaussian process model from the observations using variational autoencoders. A regularization term enforces that the predicted and inferred representation match for each timestep.

In contrast to the previous works, where the encoding from observations into latent space is learned directly, Karl et al. (2017) train a variational autoencoder to only predict the parameters of the process model and the process noise from the observations. This enforces that the learned latent state contains all information necessary to predict the next state, without relying on the observations at the next timestep.

All of the methods discussed here focus on unsupervised learning of observation and process models in systems with unknown state representation. In contrast, in our work we leverage prior knowledge about the process model and the state representation obtained from first-order principles. This enables supervised learning and is thought to improve the generalization ability of the learned parts Kloss et al. (2017). Unsupervised training by backpropagation through filtering algorithms is possible as well, as was demonstrated in (Jonschkowski & Brock, 2016).

The main conceptual difference between variational methods and learning in differentiable filters is that variational methods *learn* to perform inference in state space models by optimizing an approximate posterior distribution. Bayesian filters, on the other hand, provide fixed algorithms for approximating the posterior, which have been shown to work well in practice for many problems. Using these algorithmic priors intuitively makes learning in differentiable filters easier, but restricts the class of models that can be learned. Variational methods solve a more difficult learning problem, but can fit the training data more freely. How big this difference really is, however, depends on how the approximate posterior and the generative model in the variational approach are structured. An in-depth analysis of the effects of the two approaches on training and the learned models has, to our knowledge, not yet been attempted and would be an interesting direction for future work.

## 4  METHODS

We implement the filtering methods presented in Section 2 as recurrent neural networks in tensorflow (Abadi et al., 2015). In this section, we describe how the learnable noise models are parametrized and used in the filters. For more details about the implementation please refer to the Appendix 7.1.

### 4.1 PERCEPTION NETWORKS AND OBSERVATION NOISE

In state space models, the observation model is a generative model that predicts observations from the state $\mathbf{z}_t = h(\mathbf{x})$. In practice, it is however often hard to find such a model that directly predicts the potentially high-dimensional raw sensory signals without making strong assumptions.

We therefore use the method proposed by Haarnoja et al. (2016) and train a discriminative neural network $o$ with parameters $\mathbf{w}_o$ to preprocess the raw sensory data $\mathbf{D}$ and thus create a more compact representation of the observations $\mathbf{z} = o(\mathbf{D}, \mathbf{w}_o)$.

In our experiments, the perception network directly extracts some of the components of the state $\mathbf{x}$ from $\mathbf{D}$, such that the actual observation model $h$ becomes a simple selection operation. Besides from $\mathbf{z}$, the perception networks can also predict the diagonal entries of the observation noise covariance matrix $\mathbf{R}$.

We pretrain the perception networks for all experiments to predict $\mathbf{z}$, but not $\mathbf{R}$. Since $\mathbf{z}$ is a subset of the components of $\mathbf{x}$, this requires no additional data annotation. In experiments where $\mathbf{R}$ is learned, we initialize the prediction to reasonable values using a trainable bias, otherwise we use a fixed diagonal matrix as $\mathbf{R}$.

### 4.2 PROCESS NOISE

For learning the process noise, we consider two different conditions: constant and heteroscedastic. In all cases, we assume that the process noise at time $t$ can be described by a zero-mean Gaussian distribution with diagonal covariance matrix $\mathbf{Q}_t$. The constant noise model consists of one trainable variable $\mathbf{w}_q$ that represents the diagonal entries of $\mathbf{Q}$.

In the heteroscedastic case, the diagonal elements are predicted from the current state $\mathbf{x}_t$ and (if available) the control input $\mathbf{u}_t$, by a 3-layer MLP $g$ with weights $\mathbf{w}_g$: $\mathrm{diag}(Q) = g(\mathbf{x}_t, \mathbf{u}_t, \mathbf{w}_g)$. In the UKF and MCUKF, we predict a separate $\mathbf{Q}^i$ for every sigma point and then compute $\mathbf{Q}$ as their weighted mean.

In all variants of the Kalman Filter, the process noise enters the prediction step in the update of the covariance matrix $\mathbf{\Sigma}$ (Equations 2, 12) and influences the update step through the Kalman Gain (Equation 4). In the Particle Filter, it is used for sampling particles from the process model (Equation 13). Following Jonschkowski et al. (2018), we implement this step with the reparametrization trick (Kingma & Welling, 2013):

$$\forall \mathbf{x}_{t-1}^i \in \mathcal{X}_{t-1} \qquad \text{sample } \mathbf{n}^i \sim N(0,1) \qquad \mathbf{q}_t^i = \sqrt{\mathbf{Q}_i}\mathbf{n}^i \qquad \mathbf{x}_t^i = f(\mathbf{x}_{t-1}^i, \mathbf{u}_t, \mathbf{q}_t^i) \qquad (14)$$

### 4.3 TRAINING

We train the noise models end-to-end trough the filters using the Adam optimizer (Kingma & Ba, 2014) and backpropagation through time. The loss consists of three components, (i) the negative log likelihood of the true state given the believe, (ii) the Euclidean error between the ground truth state and the predicted mean and (iii) a regularization term on the weights of the trainable noise models.

$$L(\mathbf{l}_{0...T}, \boldsymbol{\mu}_{0...T}, \mathbf{\Sigma}_{0...T}, \mathbf{w}) =$$
$$\lambda_1 \sum_{t=0}^T \frac{1}{2}((\mathbf{l}_t - \boldsymbol{\mu}_t)^T \mathbf{\Sigma}_t^{-1}(\mathbf{l}_t - \boldsymbol{\mu}_t) + \log(|\mathbf{\Sigma}_t|)) + \lambda_2 \sum_{t=0}^T \| (\mathbf{l}_t - \boldsymbol{\mu}_t) \|_2 + \lambda_3 \| \mathbf{w} \|_2 \quad (15)$$

Here $\mathbf{l}_{0...T}$ is the ground truth state sequence, $\boldsymbol{\mu}_{0...T}$ and $\mathbf{\Sigma}_{0...T}$ denote the sequence of prediction mean and covariance respectively. $\mathbf{w}$ contains the weights of the trainable noise models (which influence the prediction of $\boldsymbol{\mu}$ and $\mathbf{\Sigma}$) like $\mathbf{w}_o$ or $\mathbf{w}_g$. The $\lambda_i$ are scaling factors that can be chosen dependent on the magnitude of the loss components.

The likelihood loss encourages the network to predict noise values that minimize the overall predicted covariance (i.e. the uncertainty about the predicted state) while at the same time penalizing high confidence predictions with large errors. In practice, we found that during learning, the models often optimized the likelihood by only increasing the predicted variance instead of minimizing

the prediction error. Therefore, we added the second component of the loss to enforce low overall prediction errors.

Both the MCUKF and the Particle Filter approximate the state by sampling and require a potentially large number of sigma points/particles for accurate prediction. During training, we have to limit the number of samples to 100, as memory consumption and computation time increase with the number of samples. For testing, we can use much higher numbers of particles/sigma points.

## 5 EXPERIMENTS

It has been shown before that using the algorithmic structure of Bayesian Filters to enable end-to-end learning is very beneficial for learning the process and observation models of the filters. Here we evaluate how end-to-end learning of *heteroscedastic noise models* affects the performance of the different filtering algorithms. These noise models quantify the accuracy of the process and observation models. For this, we test each filter under five conditions: Without learning, only learning the observation noise $\mathbf{R}$, learning only heteroscedastic process noise $\mathbf{Q}_h$ and learning both with constant or heteroscedastic process noise ($\mathbf{R} + \mathbf{Q}$, $\mathbf{R} + \mathbf{Q}_h$). As the influence of modeling the noise can depend on the task, we perform experiments on two different applications.

### 5.1 KITTI VISUAL ODOMETRY

As a first application we chose the Kitti Visual Odometry task (Geiger et al., 2012) that was also evaluated in (Haarnoja et al., 2016) and (Jonschkowski et al., 2018). The aim is to estimate the position and orientation of a driving car given a sequence of rgb images from a front facing camera and the true initial state.

The state is 5-dimensional and includes the position $\mathbf{p}$ and orientation $\theta$ of the car as well as the current linear and angular velocity $v$ and $\dot{\theta}$. As the control inputs are unknown, the estimated velocities are predicted by sampling random accelerations $a$, $\ddot{\theta}$, according to the process noise for $v$ and $\dot{\theta}$. The position and heading estimate are update by Euler integration (see Appendix 7.2.1).

While the dynamics model is simple, the challenge comes from the fact that the drivers actions are not known and the absolute position and orientation are not observable. The filters can therefore only rely on estimating the angular and linear velocity from pairs of input images to update the state, but the uncertainty about the position and heading will inevitably grow due to missing feedback.

We pretrain a neural network to extract this information from the current input image and the difference image between the current and previous one. The network architecture is the same as was used in (Haarnoja et al., 2016; Jonschkowski et al., 2018), we only replace the response normalization layers with tensorflow's standard batch normalization layers. Since both related work allowed for finetuning of the perception network trough the filter, we do the same here for better comparability of results. As in (Jonschkowski et al., 2018), we test the Particle Filter using 1000 particles and also use 1000 sigma points for the MCUKF.

The process and observation noise are initialized to the same values in every condition. For the observation noise, we look at the average error of the perception network at the end of the pretraining phase. To set the process noise, we use the ground truth standard deviation of the velocities to initialize the terms for linear and angular velocity. The terms for position and heading are initialized to identity. See the Appendix 7.2.1 for exact values.

### 5.1.1 DATA

The Kitti Visual Odometry dataset consists 11 trajectories of varying length (from 270 to over 4500 steps) with ground truth annotations for position and heading and image sequences from two different cameras collected at 10 Hz. We use the two shortest sequences for validation and perform a 9-fold cross-validation on the remaining sequences. We use both image sequences from each trajectory and further augment the data by adding the mirrored sequences as well. For training, we extract non-overlapping sequences of length 50 with a different random starting point for each image-sequence. The sequences for validation and testing consist of 100 timesteps.

| | No learning | $\mathbf{R}$ | $\mathbf{Q}_h$ | $\mathbf{R + Q}$ | $\mathbf{R + Q}_h$ |
|---|---|---|---|---|---|
| Translational error [m/m] | | | | | |
| EKF | $\mathbf{0.22 \pm 0.13}$ | $\mathbf{0.19 \pm 0.13}$ | $\mathbf{0.19 \pm 0.11}$ | $\mathbf{0.20 \pm 0.13}$ | $\mathbf{0.19 \pm 0.12}$ |
| UKF | $0.39 \pm 0.14$ | $0.29 \pm 0.15$ | $0.41 \pm 0.17$ | $0.26 \pm 0.14$ | $0.24 \pm 0.16$ |
| MCUKF | $0.99 \pm 0.04$ | $1.01 \pm 0.02$ | $0.36 \pm 0.14$ | $0.76 \pm 0.09$ | $0.44 \pm 0.25$ |
| PF | $0.81 \pm 0.95$ | $0.73 \pm 0.40$ | $0.40 \pm 0.25$ | $0.26 \pm 0.17$ | $0.3 \pm 0.22$ |
| Rotational error [deg/m] | | | | | |
| EKF | $\mathbf{0.11 \pm 0.10}$ | $\mathbf{0.17 \pm 0.19}$ | $\mathbf{0.10 \pm 0.10}$ | $\mathbf{0.11 \pm 0.08}$ | $\mathbf{0.10 \pm 0.13}$ |
| UKF | $0.26 \pm 0.08$ | $0.29 \pm 0.35$ | $0.21 \pm 0.11$ | $0.14 \pm 0.06$ | $0.16 \pm 0.12$ |
| MCUKF | $1.17 \pm 0.92$ | $1.24 \pm 0.56$ | $0.30 \pm 0.28$ | $0.48 \pm 0.42$ | $0.27 \pm 0.20$ |
| PF | $0.95 \pm 0.89$ | $0.93 \pm 0.85$ | $0.29 \pm 0.26$ | $0.86 \pm 1.31$ | $0.20 \pm 0.20$ |

Table 1: Kitti Visual Odometry task. Evaluation of four non-linear filters under five different noise learning conditions: No learning, learning constant observation noise $\mathbf{R}$, learning heteroscedastic process noise $\mathbf{Q}_h$, learning constant observation and process noise $\mathbf{R + Q}$, learning constant observation noise and heteroscedastic process noise $\mathbf{R + Q}_h$. In each condition, the perception network was pretrained offline and finetuned through the filters. We evaluate the models on different trajectories with 100 timesteps. As in (Jonschkowski et al., 2018; Haarnoja et al., 2016) we report mean and std of the end-point- error in position and orientation normalized by the distance between start and end point.

### 5.1.2 RESULTS

Table 1 contains the average normalized end-point-errors for the different filters and noise learning conditions. On this task, the EKF outperforms the other filters even without learning the noise models and does not gain a lot from leaning them.

The Particle Filter as well as the MCUKF perform badly without learning or when training the observation noise $\mathbf{R}$ alone. While learning a constant process noise $\mathbf{Q}$ improved their results, learning a heteroscedastic process noise model lead to much bigger improvements for the MCUKF and for the PF when predicting the heading of the car.

This does not necessarily mean that the task follows a heteroscedastic noise model, especially since the EKF and UKF do not show big differences between constant and heteroscedastic noise. Instead, it seems like the heteroscedastic process noise model facilitates the training process: When the process noise is trained with a heteroscedastic process noise model, we observe that it quickly converges towards zero for position and orientation, which is the best choice for this task. In the constant noise setting, this convergence is much slower and the models do not fully converge during the training.

While our EKF results are close to those reported in Haarnoja et al. (2016) (translation: $0.21\frac{m}{m}$, rotation: $0.08\frac{deg}{m}$), our results for the Particle Filter are notably worse than the results reported by Jonschkowski et al. (2018) (translation: $0.15\frac{m}{m}$, rotation: $0.05\frac{deg}{m}$). This could be due to differences in the implementation of the observation model (Jonschkowski et al. (2018) use a model that directly predicts the likelihood of each particle instead of a distribution over velocities), the different initial values for the process noise or the soft resampling we use (see Appendix 7.1.3).

For this particular task, the MCUKF turns out to be a bad choice: Without learning a suitable process noise model, it mostly fails to predict any movement of the car. This is caused by high uncertainty about the orientation of the car, both due to the bad initialization of the process noise and the accumulating uncertainty during tracking: If the sampled sigma points are too different in estimated orientation, their movement cancels each other out when calculating the mean. The standard UKF performs better, because of the symmetry in the sigma point construction (see Eq. 11) and because it keeps the previous mean as a sigma point that is weighted higher than the remaining points and thus enforces movement in the correct direction.

In general, it is not surprising that the EKF performs best on this task: First, the process model is smooth and not highly non-linear, such that the EKF provides a good approximation of the posterior. Second, the main difference to the other filters is that the PF and the UKF variants generate additional

uncertainty about the position and heading of the car by sampling particles or constructing sigma points. This uncertainty would usually be resolved by observations, such that more weight can be given to the particles or sigma points that are closer to the true state. In visual odometry, however, there are no observations of heading and position and there is thus nothing to gain from exploring values that deviate from the estimated mean.

## 5.2 PLANAR PUSHING

In the visual odometry problem, the main challenges were perception and dealing with the inevitably increasing uncertainty. Our second experiment in contrast addresses a task with more complex dynamics: quasi-static planar pushing. Apart from having non-linear and discontinuous dynamics (when the pusher makes or breaks contact with the object), Bauza & Rodriguez (2017) also showed that the noise in the system can be best captured by a heteroscedastic noise model.

The state we try to estimate has 10 dimensions: the 2d position $\mathbf{p}$ and orientation $\theta$ of the object, two friction-related parameters $l$ and $m$, the 2d contact point between pusher and object $\mathbf{r}$ and the normal to the object's surface there $\mathbf{n}$ as well as a variable $s$ that indicates if the pusher is in contact with the object or not.

For predicting the next state, we use an analytical model of quasi-static planar pushing (Lynch et al., 1992; Kloss et al., 2017). It predicts the linear and angular velocity of the object $(\mathbf{v}, \omega)$ given the pusher velocity $\mathbf{u}$ and the current state. Details can be found in the Appendix 7.2.2.

We use coordinate images (like a depth image, but with all 3 coordinates as channels) of the scene at time $t-1$ and $t$ as input, and train a neural network to extract the position of the object, the contact point and normal as well as if the pusher is in contact with the object or not. Besides from the friction-related parameters, the orientation of the object, $\theta_t$, is the only state component that cannot be estimated directly from the input images. As absolute orientation of an object is not defined (without giving a reference for each object), we cannot extract it from the images. Instead, we train the network to observe the change in orientation $\omega$ between the two images (up to symmetries).

In contrast to the visual odometry task in the previous experiment, we do not assume that the initial state is correct. All models are thus evaluated on five different initial conditions with varying error and we report the average error and standard deviation across these five setting. We also do not finetune the perception model in this experiment. We again use 100 sigma points or particles during training for the MCUKF and PF. During test-time, the particle filter uses 1000 particles while we limit the MCUKF to 500 sigma points.

### 5.2.1 DATA

The MIT Push dataset (Yu et al., 2016) consist of more than a million real robot push sequences to eleven different objects on four different surface materials. For each sequence the original dataset contains the object position, the position of the pusher as well as force recordings. We use the tools described by Kloss et al. (2017) to get additional annotations for the remaining state components and for rendering depth images. In contrast to (Kloss et al., 2017) our images also show the robot arm and are taken from a more realistic camera angle.

We use data from pushes with a velocity of 50 $\frac{mm}{s}$ and render images with a frequency of 18 Hz. This results in very short sequences of about 15 images per push. We extend these sequences to 100 steps by chaining multiple pushes and adding in between pusher movement when necessary. We use subsequences of ten steps for training and the full 100 steps for testing.

### 5.2.2 RESULTS

**Without learning** In the first two columns of Table 2, we compare the tracking performance of the different filters without learning any of the noise models. For the first column, we set the diagonal values of $\mathbf{Q}$ to 0.01 and those of $\mathbf{R}$ to 100 such that the filters place too high confidence in the process model and too low confidence in the observations. In the second condition, we used the average prediction error of the analytical model and the preprocessing network on the ground truth data to set $\mathbf{Q}$ and $\mathbf{R}$ to realistic values.

| | No learning 1 | No learning 2 | **R** | **$Q_h$** | **R + Q** | **R + $Q_h$** |
|---|---|---|---|---|---|---|
| Translational error [mm] | | | | | | |
| EKF | $11.8 \pm 0.54$ | $3.9 \pm 0.02$ | $\mathbf{3.7 \pm 0.01}$ | $3.9 \pm 0.01$ | $3.8 \pm 0.01$ | $3.9 \pm 0.02$ |
| UKF | $9.3 \pm 0.31$ | $\mathbf{3.8 \pm 0.01}$ | $3.8 \pm 0.02$ | $3.8 \pm 0.02$ | $3.8 \pm 0.01$ | $3.9 \pm 0.003$ |
| MCUKF | $\mathbf{9.2 \pm 0.33}$ | $\mathbf{3.8 \pm 0.01}$ | $\mathbf{3.7 \pm 0.01}$ | $3.8 \pm 0.01$ | $\mathbf{3.7 \pm 0.01}$ | $3.8 \pm 0.01$ |
| PF | $56.5 \pm 0.11$ | $7.4 \pm 0.30$ | $20.2 \pm 0.62$ | $\mathbf{3.3 \pm 0.23}$ | $7.0 \pm 0.21$ | $\mathbf{3.0 \pm 0.20}$ |
| Rotational error [deg] | | | | | | |
| EKF | $\mathbf{18.9 \pm 0.57}$ | $\mathbf{9.3 \pm 0.3}$ | $9.9 \pm 0.21$ | $9.4 \pm 0.39$ | $8.4 \pm 0.24$ | $9.3 \pm 0.33$ |
| UKF | $20.6 \pm 1.11$ | $9.4 \pm 0.28$ | $10.8 \pm 0.22$ | $9.34 \pm 0.26$ | $9.5 \pm 0.17$ | $\mathbf{6.1 \pm 0.14}$ |
| MCUKF | $21.4 \pm 1.4$ | $10.1 \pm 0.43$ | $\mathbf{9.5 \pm 0.38}$ | $\mathbf{7.6 \pm 0.26}$ | $\mathbf{8.4 \pm 0.2}$ | $6.5 \pm 0.21$ |
| PF | $28.4 \pm 0.07$ | $21.1 \pm 1.1$ | $16.4 \pm 0.46$ | $8.8 \pm 0.18$ | $12.2 \pm 0.45$ | $10.1 \pm 0.37$ |

Table 2: Planar Pushing task. Evaluation of four non-linear filters under five different noise learning conditions: No learning 1 (with unrealistic noise), No learning 2 (with realistic noise) , learning constant observation noise $\mathbf{R}$, learning heteroscedastic process noise $\mathbf{Q}_h$, learning constant observation and process noise $\mathbf{R} + \mathbf{Q}$, learning constant observation noise and heteroscedastic process noise $\mathbf{R} + \mathbf{Q}_h$. Mean and standard deviation of tracking errors on the planar pushing task averaged over five different initial conditions. Tracking errors are mean squared error in position and orientation of the object averaged over all timesteps in the sequence.

While all filters perform worse on the unrealistic noise setting, the Particle Filter is affected the most. This is presumably because without well-tuned noise models, it samples many particles far away from the true state and cannot discriminate well between likely and unlikely particles given the observations.

**Learning the noise models**   The remaining columns of Table 2 show the results when learning the different combinations of noise models. The process and observation noise are initialized to the realistic values from the no learning setting for every condition. We can see that the performance of the Extended Kalman Filter again remains mostly constant over all conditions and also does not improve much over the model with well-tuned noise.

Both the UKF and the MCUKF do not show much difference for tracking the position of the object. We see a slightly improved performance for tracking the orientation of the object when a constant process noise model is trained and a stronger improvement with the heteroscedastic $\mathbf{Q}_h$. This is consistent with the results in the previous experiment, as the orientation of the object can again not be observed directly and it is thus not desirable to vary it much when creating the sigma points. Overall, the traditional UKF with trained $\mathbf{R}$ and heteroscedastic $\mathbf{Q}$ performs best, but the MCUKF is similar and could potentially perform better if more sigma points were sampled.

In this experiment, the Particle Filter profits most from learning: In the two conditions with heteroscedastic process noise, its tracking performance improves dramatically and even outperforms the other filters on the position metric. The improvement over the untrained setting is much smaller when $\mathbf{Q}$ is constrained to be constant. Why is learning a heteroscedastic process noise model so important for the PF? We believe that learning a separate $\mathbf{Q}$ for each particle helps the filter to steer the particle set towards more likely regions of the state space. It can for example get rid of particles that encode a state configuration that is not physically plausible and will therefore lead to a bad prediction from the analytical model by sampling higher noise and thus decreasing the likelihood of the particle.

Training the observation noise $\mathbf{R}$ did not have a very big effect in this experiment, but inspecting the learned diagonal values showed that all filters learned to predict higher uncertainty for the $y$ coordinate of positions, which makes sense as the $y$ axis of the world frame points towards the background of the image and perspective transform thus reduces the accuracy in this direction. In contrast to the results in (Haarnoja et al., 2016), we did not see any evidence that the heterostochasticity of the observation noise was helpful. This can probably be explained by the absence of complete occlusions of the object in our dataset. We could also not identify any other common feature of scenes for which our prediction model produced high prediction errors. It is therefore likely that a constant observation noise model would have been sufficient in this setting.

## 6 CONCLUSIONS

We proposed to optimize the process and observation noise for Bayesian Filters through end-to-end training and evaluated the method with different filtering algorithms and on two robotic applications. Our experiments showed that learning the process noise is especially important for filters that sample around the mean estimate of the state, like the Particle Filter but also the Unscented Kalman Filters. The Extended Kalman Filter in contrast proved to be most robust to suboptimal choices of the noise models. While this makes it a good choice for problems with simple and smooth dynamics, our experiments on the pushing task demonstrated that the (optimized) Unscented Filters can perform better on problems with more complex and even discontinuous dynamics.

Training a state-dependent process noise model instead of a constant one improves the prediction accuracy for dynamic systems that are expected to have heteroscedastic noise. In our experiments, it also facilitated learning in general and lead to faster convergence of the models.

We also used a heteroscedastic observation noise model in all our experiments. But different from the results in (Haarnoja et al., 2016), we could not see a large benefit from it: Inspection on the pushing task showed that larger errors in the prediction of the preprocessing networks were not associated with higher observation noise. Identifying inputs that will lead to bad predictions is a difficult task if no obvious problems like occlusions are present to explain such outliers. Developing better methods for communicating uncertainty about the predictions of a neural network would thus be an impotent next step to further improve the performance of differentiable Bayesian Filters.

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

## 7 Appendix

### 7.1 Implementation Details

#### 7.1.1 EKF

The basic steps of the Extended Kalman Filter can be directly implemented in Tensorflow without any modifications. The only aspect of interest is how to compute the Jacobians of the process and observation model. Tensorflow implements auto differentiation, but has (as of now) no native support for computing Jacobians. While it can be done, it requires looping over the dimensions of the differentiated variable one by one, which we found to be relatively slow, especially during graph-construction. We therefore recommend to manually derive the Jacobians where applicable.

#### 7.1.2 UKF and MCUKF

Like for the EKF, implementing the prediction and update step of the UKF in tensorflow is straight forward. For constructing the sigma points, it is necessary to compute the matrix square root of the estimated covariance $\Sigma$. This is commonly done using the Cholesky Decomposition, which is also available in tensorflow. In practice, the Cholesky decomposition however often failed. Instead, we used the more robust singular value decomposition.

For the MCUKF, we sample the sigma points from a Gaussian distribution with the same mean and covariance as the current estimate using tensorflow's distribution tools. Internally, this also relies on the Cholesky decomposition and thus requires $\Sigma$ to be positive semidefinite at all times.

#### 7.1.3 PF

Our particle Filter implementation is very similar to the variant proposed by Jonschkowski et al. (2018) that is available online[1]. We combine it with the differentiable resampling technique proposed by Karkus et al. (2018) to enable backpropagation through the weights.

Another difference is that we do not train a network to directly predict the likelihood of an observation given a particle. Instead, we use the same preprocessing network as for the other filtering types, which outputs the observations $\mathbf{z}$ and the estimated covariance matrix of the observation model $\mathbf{R}$. Given these, we compute the probability of $\mathbf{z}$ under a gaussian distribution defined by the predicted observations for each particle and $\mathbf{R}$. This approach might be more challenging to train (as the likelihoods become very small if the observation noise is too low) but allows for a better comparison with the other filters.

#### 7.1.4 Stability

A particular difficulty in training differentiable filters in tensorflow is to ensure that the estimated covariance matrices are positive semidefinite at any time, even if the filters diverge. This ensures for example that they can be inverted for computing likelihoods or the Kalman Gain, which will otherwise result in an error that stops the training. We employ the method described in Higham (1988) to reset the covariance matrices to the nearest positive semidefinite matrix after every iteration.

---

[1] https://github.com/tu-rbo/differentiable-particle-filters

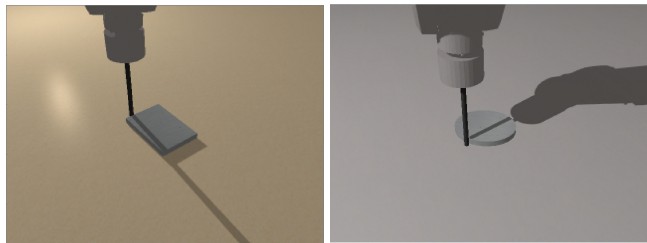

Figure 1: Examples of rendered rgb images for the pushing task.

## 7.2 EXPERIMENTAL DETAILS

### 7.2.1 KITTI VISUAL ODOMETRY TASK

The process model for the visual odometry task is defined as

$$\begin{pmatrix} p_x \\ p_y \end{pmatrix}_t = \begin{pmatrix} p_x \\ p_y \end{pmatrix}_{t-1} + \Delta t v_{t-1} \begin{pmatrix} \sin(\theta_{t-1}) \\ \cos(\theta_{t-1}) \end{pmatrix} \qquad v_t = v_{t-1} + \Delta t a_t$$

$$\theta_t = \theta_{t-1} + \Delta t \dot{\theta}_{t-1} \qquad \dot{\theta}_t = \dot{\theta}_{t-1} + \Delta t \ddot{\theta}_t$$

For the architecture of the preprocessing network, we refer to Haarnoja et al. (2016). We initialize the process noise with diagonal values of

$$\text{diag}(\mathbf{Q}) = (1 \quad 1 \quad 1 \quad 11. \quad 0.0225)$$

and the observation noise with

$$\text{diag}(\mathbf{R}) = (4. \quad 1)$$

### 7.2.2 PLANAR PUSHING TASK

**Data**   Figure 1 shows two examples of the rendered images we use in the pushing task. While we actually use coordinate images as input, we show rgb images here for better visibility.

**Preprocessing Network**   The architecture of the preprocessing network that infers $\mathbf{z}$ from the raw input images is shown in Figure 2. It is similar to the network described in (Kloss et al., 2017) for inferring object position $\mathbf{p}$, contact point $\mathbf{r}$, contact normal $\mathbf{n}$ and the contact indicator $s$ from the scene. We add the left part that computes $\omega$, the difference in object rotation between the current and the previous image. For this, we extract patches around the predicted object position in both images and feed both into a convolutional and fully-connected network to infer $\omega$.

**Process Model**   Given the output of the analytical model $(\mathbf{v}_t, \omega_t)$, we formulate the process model $f(\mathbf{x}_t, \mathbf{u}_t)$ as

$$\begin{aligned}
\mathbf{p}_{t+1} &= \mathbf{p}_t + \mathbf{v}_t & \mathbf{r}_{t+1} &= \mathbf{r}_t + \mathbf{u}_t \\
\theta_{t+1} &= \theta_t + \omega_t & \mathbf{n}_{t+1} &= \mathbf{R}(\omega_t)\mathbf{n}_t \\
l_{t+1} &= l_t & s_{t+1} &= s_t \\
m_{t+1} &= m_t
\end{aligned}$$

Here, we make the simplifying assumption that the pusher will not make or break contact and that $s$ is thus constant. To predict the next contact point, we update it with the movement of the pusher. The accuracy of this prediction is bounded by the radius of the pusher, which is rather small in our case. For predicting the next normal at the contact point, we assume that the position of the contact point on the object does not change and the normal thus remains constant in the object coordinate frame. Given this assumption, the only thing we need to do is to adapt the orientation of the normal to the rotation of the object, where $\mathbf{R}(\omega_t)$ denotes a rotation matrix that rotates $\mathbf{n}$ by $\omega_t$.

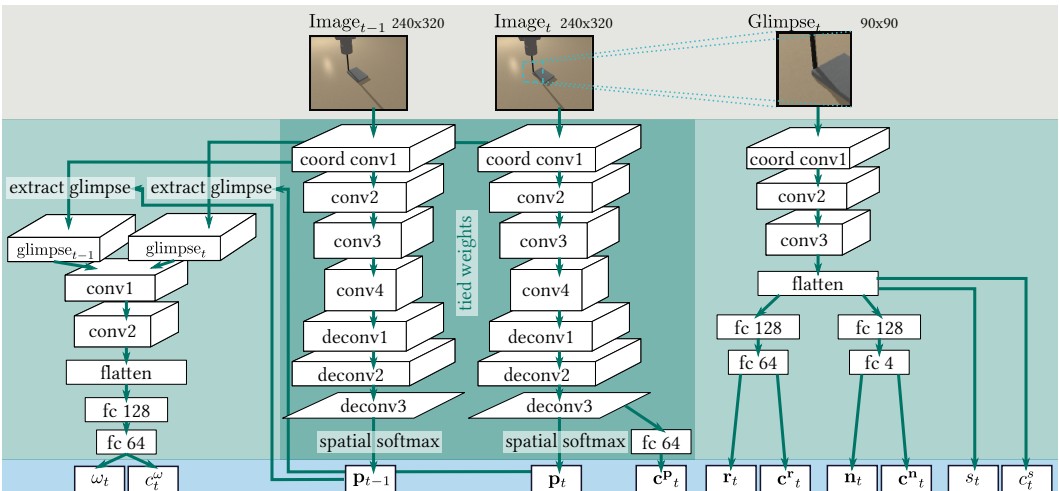

Figure 2: The preprocessing network for extracting the observations for the pushing task and their associated uncertainty from two input corrdinate images.

Using empirical data from the analytical model and the preprocessing model, we initialize the process noise with diagonal values of

$$\mathrm{diag}(\mathbf{Q}) = (4 \quad 4 \quad 4 \quad 0.01 \quad 0.01 \quad 1. \quad 1. \quad 0.0625 \quad 0.0625 \quad 0.5)$$

and the observation noise with

$$\mathrm{diag}(\mathbf{R}) = (16. \quad 16. \quad 0.5625 \quad 4. \quad 4. \quad 0.25 \quad 0.25 \quad 0.09)$$

