# OpenReview forum: "On Learning Heteroscedastic Noise Models within Differentiable Bayes Filters"
_ICLR.cc/2019/Conference_

### Official Review · AnonReviewer1 · 2018-11-01
**Good exploration of optimizing Bayesian filter noise variance through back propagation, but with incomplete results**

**Rating:** 6
**Confidence:** 4

**Review:**

This is a well written paper which proposes to learn heteroscedastic noise models from data by optimizing the prediction likelihood end-to-end through differentiable Bayesian Filters. In addition to existing Bayesian filters, the paper also proposes two different versions of the [differentiable] Unscented Kalman Filter. Performance of the different filters and noise models is evaluated on two real-world robotic problems: Visual Odometry and visual tracking of an object pushed by the robot.
While the general idea of learning the noise variances through backpropagation are straightforward extensions of existing work on differential Bayesian filters, the questions that the paper explores are important to make end-to-end learning of Bayesian filter more common. The results will help future research select the correct differential filter for their use case, and insight in potential benefits (or lack thereof) by learning heteroscedastic or homoscedastic process noise, and/or observation noise.
A downside is that the paper does not further explore how to weigh different loss terms which are apparently important to successfully train such models. Also unfortunate is the footnote which states that the current results are incomplete and will be updated, hence as a reviewer I am not sure which results and conclusions are valid right now.


Pros:
+ clearly written
+ useful experiments for those seeking to select a differential Bayesian filter, and learning (heteroscedastic) noise from data.
+ experiments on real-world use cases rather than toy problems

Cons:
- Incomplete experiments according to footnote, thus results and conclusions might change after this review.
- Unclear what the effect of the selected process / observation model is on the learned noise


Below are more detailed comments and questions:
* p6. Footnote: "due to time constraints, ..., results will be updated" Is this acceptable? I have never seen such a notice when reviewing. So, are the current results on a single fold? Will the numbers in the tables, or the conclusions change after this review?
* If I understand correctly, the paper 'only' focuses on learning the heteroscedastic noise variance, but assumes that the deterministic non-linear parts of the process and observation models are fixed. I did not find this very clearly stated in the paper, though at least the Appendix explicitly states the used functions for the process models.
* I would have liked to see in the paper more explanation on how the process and observations models were selected and validated  in the experiments, since I expect that the validity of these functions affects the learned noise variances. Since the noise needs to account for the inaccuracies in the deterministic models, would the choice for these functions not impact your conclusions? And, would it or would it not be possible to learn both these deterministic models and the noise jointly from the training data?
* Is it possible to add priors on Q and R parameters for Bayesian treatment of learning model parameters? I can imagine that priors can guide the optimization to either adjust more of the Q or more of the R variance to improve the likelihood.

* Section 1:
	* "Our experiments show that ... " This may be a matter of taste, but I did not expect to see the main conclusions already in the introduction. They should appear in the abstract to help out the quick reader. In the introduction, it appears as if you are talking about some separate preliminary experiments, and which you base some conclusions that will be used in the remainder of this paper.

* Section 3:
	* So, mostly empirical study, since heteroscedastic noise models were already used?
	* "Previous work evaluated ... " please add citations

* Section 4.1:
	* "train a discriminative neural network o with parameters wo to preprocess the raw sensory data D and thus create a more compact representation of the observations z = o(D;wo)." At this point in the paper, I don't understand this. How is z learned, via supervised learning (what is the target value for z)? Or is z some latent representation that is jointly optimized with the filters? This only became somewhat clearer in Sec. 5.2 on p.8 where it states that "We ... train a neural network to extract the position of the object, the contact point and normal as well as ...". So if I understand correctly, the function o for z = o(D) is thus learned offline w.r.t. some designed observation variables for which GT is available (from manual annotations?).

* Section 4.2:
	* "we predict a separate Qi for every sigma point and then compute Q as the weighted mean" → So, separate parameters w_g for each sigma point i, or is a single learned non-linear function applied to all points?

* Section 4.3:
	* Equation 14: inconsistent use of boldface script: should use bold sigma_t, and bold l_t ?
	* "In practice, we found that during learning ... by only increasing the predicted variance" →  This is an interesting observation, which I would have liked to see explored more. I understand that term (ii) is needed to guide the learning processes, but in the end wouldn't we want to optimize the actual likelihood? So, could you (after the loss with (ii) converged) reduce \lambda_2 to zero to properly optimize only the log likelihood without guidance from a good initial state? Or is it not possible to reliably optimize the likelihood via back-propagation at all from some reason?

* Section 5.1.1
	* "... of varying length (from 270 to over 4500 steps) ..." it would be good to mention the fps, to get understand to what real-world time horizons 50 / 100 frames correspond.

* Section 5.1.2:
	* Table 1: How are the parameters of the filters in the "no learning" column obtained? Are these tuned in some other way, or taken form existing implementations? Also, can you clarify if the 'no learning' parameters served as the initial condition for the learning approaches?
	* Table 1, first row column Q+R: "0.2" → Is there a missing zero here, i.e. "0.20"? Otherwise, the precision of reported results in this table is not consistent. Hard to say: is the mean of R+Q 0.2, and slightly lower than R+Qh, or could it be as high as 0.24 ?
	* "learning a heteroscedastic process noise model leads to big improvements and makes the filters competitive with the EKF". Results for EKF still appear significantly better than the novel UKF, and even the PF (especially rotational error).

* Section 6:
	* "Large outliers in the prediction of the preprocessing networks were not associated with higher observation noise." I don't see on what presented results these conclusions were drawn, as this is the first time the word "outlier" is mentioned in the paper. Outliers seem indeed important, as they contradict the typical assumptions e.g. of Gaussian noise, so it would be useful to clarify how the proposed techniques handle such outliers.

---

> ### Author Response · Authors · 2018-11-24
> **Response (1/3)**
>
> Thanks a lot for the positive and detailed review. The suggested experiments about weighting
> the different loss terms are a very interesting direction which we would like to explore further. However, due to the rather long training time of such models, we can not include such experiments within the rebuttal period.
>
>
> Replies to your questions:
>
> * p6. Footnote [...] So, are the current results on a single fold? Will the numbers in the tables, or the conclusions change after this review?
>
> Reply:
> The footnote was poorly worded, sorry for this. The results were from a 4-fold cross-validation instead of the full 9-fold cross validation that can be done when using all sequences.
>
> We updated the results. The reason for the long delay is that we decided to allow for finetuning of the perception model for the kitti experiments to improve the comparability to related work, which required rerunning the experiments. This does not affect the conclusion, but improves the overall results.
>
> * If I understand correctly, the paper 'only' focuses on learning the heteroscedastic noise variance, but assumes that the deterministic non-linear parts of the process and observation models are fixed.
> I did not find this very clearly stated in the paper, though at least the Appendix explicitly states the used functions for the process models.
>
> Reply:
> This is correct. We tried to make this clearer in the introduction and the experimental section.
>
> * I would have liked to see in the paper more explanation on how the process and observations models were selected and validated [...] Since the noise needs to account for the inaccuracies in the deterministic models, would the choice for these functions not impact your conclusions? And, would it or would it not be possible to learn both these deterministic models and the noise jointly from the training data?
>
> Reply:
> The perception networks were pretrained on ground truth data and selected to have a small
> prediction error. For the kitti-task, we followed the implementation of (Jonschkowski et al., 2018) for the prediction network architecture as well as the process model for comparability.
> On the pushing task, we selected an architecture similar to (Kloss et al., 2017) for the perception network. The analytical model we used as a process model was also evaluated in this work.
>
> We agree that the choice of those models also impacts the absolute results, as does the specific task at hand (as can be seen from the difference between the kitti and the pushing results). We however believe that the chosen models capture the real process close enough to not influence the conclusion. How the method behaves with worse models would be an interesting question for future work.
>
> Concerning the question if all could be learned jointly: For the kitti task, (Jonschkowski et al., 2018) demonstrated that it is possible to learn everything jointly from scratch. However, they reported better results when pretraining the models first and then finetuning through the filter.
> We did not try it on the pushing task, which we expect to be more difficult to learn from scratch, as the state and observations have more dimensions. In general, the results found in e.g.
> (Kloss et al., 2017) and (Sahoo et al., 2018)  also suggest that learning process models might not be desirable if a suitable analytical model is available, due to limited generalization abilities of neural networks.
>
> * Is it possible to add priors on Q and R parameters for Bayesian treatment of learning model parameters? I can imagine that priors can guide the optimization to either adjust more of the Q or more of the R variance to improve the likelihood.
>
> Reply:
> Adding priors for a bayesian treatment of the noise models is as far as we know not trivially
> possible within the framework of filtering. For this, a variational approach might be better suited.
> In our approach, we do however supply a form of prior on the value of the covariance matrices in the form of an initialization bias.
>
>
> * Section 3: So, mostly empirical study, since heteroscedastic noise models were already used?
>
> Reply:
> True, it is mostly an ablation study of concepts that were already proposed in other work but not evaluated in isolation. However, in contrast to previous work, we actually evaluate the heteroscedastic noise model on a task that is known to follow heteroscedastic process noise (pushing). In addition, we proposed the two differentiable UKF variants and did a comparison of the different filtering techniques on two different tasks.

---

> ### Author Response · Authors · 2018-11-24
> **Response (2/3)**
>
> * Section 4.1: [...] How is z learned, via supervised learning (what is the target value for z)?
> Or is z some latent representation that is jointly optimized with the filters? [..]
> So if I understand correctly, the function o for z = o(D) is thus learned offline w.r.t. some designed observation variables for which GT is available (from manual annotations?).
>
> Reply:
> Yes, z is a subset of the quantities in the state x, all those which can be observed from the input images. This means that we have ground truth values for training z in a supervised way, without providing any additional annotations beyond the ground truth states. We tried to make this more clear in the updated version.
>
> * Section 4.2: "we predict a separate Qi for every sigma point and then compute Q as the weighted mean"
> So, separate parameters w_g for each sigma point i, or is a single learned non-linear function applied to all points?
>
> Reply:
> The network for predicting the Qi is the same for every sigma point (shared weights w_g). However, in the UKF the sigma points have associated weights that are used for computing mean and covariance of the approximate posterior. We use these "per sigma point" weights to compute one Q from the different Qi. (In the MCUKF, those weights are uniform)
>
> * Section 4.3: […]
> This is an interesting observation, which I would have liked to see explored more. I understand that term (ii) is needed to guide the learning processes, but in the end wouldn't we want to optimize the actual likelihood? So, could you (after the loss with (ii) converged) reduce \lambda_2 to zero to properly optimize only the log likelihood without guidance from a good initial state? Or is it not possible to reliably optimize the likelihood via back-propagation at all from some reason?
>
> Reply:
> As said above, we agree that more experiments on this would be very interesting. Our intuition was that once the training is in a "good" state, term (ii) would be small, such that the likelihood would be the main driver for the training even without changing the weighting of the loss terms.
>
> * Section 5.1.1 [...] it would be good to mention the fps, to get understand to what real-world time horizons 50 / 100 frames correspond.
>
> Reply:
> For the kitti-dataset, images were recorded with a frequency of 10Hz, so 100 frames correspond to 10 seconds of driving. We included this information in the updated version.
>
> * Section 5.1.2:
> 	* Table 1: How are the parameters of the filters in the "no learning" column obtained? Are these tuned in some other way, or taken form existing implementations? Also, can you clarify if the 'no learning' parameters served as the initial condition for the learning approaches?
>
> Reply:
> The noise values for linear and angular velocity (the 2 last components) were chosen based on the ground-truth standard deviation of one set of the kitti data.
> For the position and heading, we could think of no obvious way of choosing the values based on data and thus set them to identity, which is a common choice for the first values to try out.
> And yes, we used these values as initial conditions in the learning settings, by adding a bias term with a strong weight decay.
>
> 	* Table 1, first row column Q+R: "0.2" → Is there a missing zero here, i.e. "0.20"? Otherwise, the precision of reported results in this table is not consistent.
>
> Reply:
> Sorry for the confusion. Yes, the notation was meant to convey 0.20, we changed it.
>
> 	* "learning a heteroscedastic process noise model leads to big improvements and makes the filters competitive with the EKF". Results for EKF still appear significantly better than the novel UKF, and even the PF (especially rotational error).
>
> Reply:
> True, there is also no reason why the (MC)UKF or PF should have any advantage on the kitti task: For once, the process model is not "heavily" nonlinear, such that the EKF is a good choice.
> The main problem here is that the sampling and the sigma points generate additional uncertainty about the state-estimate. Usually, this would be resolved by observations, such that more weight can be given to the particles/sigma points that are closer to the true state. But in the visual odometry task, there are no observations of heading and position. Varying those parts of the state thus only adds uncertainty, but does not help.  We updated the results section to better explain this.
>
> However, (Jonschkowski et al., 2018) obtained much better results with their version of the particle filter. The main difference to our version is that their perception model directly predicts likelihoods from the observations and the particles. This approach seems to allow for a better weighting of the particles. (We chose to use the same perception model as for the other filters to allow for better comparability between them)

---

> ### Author Response · Authors · 2018-11-24
> **Response (3 /3)**
>
> * Section 6:
> 	* "Large outliers in the prediction of the preprocessing networks were not associated with higher observation noise." I don't see on what presented results these conclusions were drawn, as this is the first time the word "outlier" is mentioned in the paper. Outliers seem indeed important, as they contradict the typical assumptions e.g. of Gaussian noise, so it would be useful to clarify how the proposed techniques handle such outliers.
>
> Reply:
> Agreed, we tried to make it clearer what we meant by outliers.
> In this case, outliers were mostly meant to mean "unusually bad predictions" especially of the object position in the pushing task. An important point here is that on the pushing task there is no structural explanation for the bad predictions (such as for example occlusions). Therefore we do not think that they actually violate a gaussian assumption about the observation noise.
>
> On the question of how the method handles outliers:
> The idea behind using a heteroscedastic noise model is that it allows to assign different levels of noise to different inputs. For example, if the object is occluded in the image, a high observation noise can be predicted. This flags the observations in this timestep as unreliable, such that the filters rely more on the process model prediction.
> Such outliers would indeed violate a global gaussian assumption about the observation noise. To alleviate this, our method instead learns input-dependent "local" noise distributions. This allows it to capture e.g. the noise in the unoccluded case with one distribution and the prediction errors in the case of occlusion with another one.
>
>
> References:
>
> Rico Jonschkowski, Divyam Rastogi, and Oliver Brock. Differentiable particle filters: End-to-end learning with algorithmic priors. In Proceedings of Robotics: Science and Systems, Pittsburgh, USA, 2018.
>
> Alina Kloss, Stefan Schaal, and Jeannette Bohg. Combining learned and analytical models for
> predicting action effects. arXiv preprint arXiv:1710.04102, 2017.
>
> Subham Sahoo, Christoph Lampert and Georg Martius. Learning Equations for Extrapolation and Control. In Proceedings of the 35th International Conference on Machine Learning, PMLR 80:4442-4450, 2018.

---

### Official Review · AnonReviewer3 · 2018-11-02
**Small novelty with insufficient novelty**

**Rating:** 4
**Confidence:** 4

**Review:**

This paper presents a method to learn and use state and observation dependent noise in traditional Bayesian filtering algorithms.  For observation noise, the approach consists of constructing a neural network model which takes as input the raw observation data and produces a compact representation and an associated (diagonal) covariance.  Similarly for state process noise, a network predicts the (diagonal) covariance of the temporal process given the current state.

The paper notes that these noise models can be trained end-to-end by instantiating an (approximate) Bayesian filter.  In particular, they explore the use of a Kalman Filteer, Extended Kalman Filter, (Monte Carlo and regular) Unscented Kalman Filter and a Particle Filter.

The technique is applied to two different tasks, visual odometry on the KITTI dataset and a "planar pushing" task.  The results show that the addition of a learned noise model made no significant difference on the KITTI dataset, with the EKF without learning performing as well as any of the other variations.  The planar pushing task has a higher dimensional state space and more challenging noise dynamics.  In that case some gains are seen with learning.

Overall the contribution of this paper seems small and the experimenal results insufficient.  The observation that gradient based training can be done through a Bayesian filter, as the paper pointed out, was developed elsewhere.  Extending that to a more complex noise model seems like a rather small contribution.  Indeed, the observational noise component was not found to have a significant or consistent impact and hence only the process noise is particularly notable.  Further, at least one obvious and important baseline was missing.  Specifically, process noise models could be trained independently by simply maximizing the likelihood of the next predicted state.  It's not clear that there's a significant benefit to training the model end-to-end in this case.  There may well be, but that is something that should be demonstrated.

A number of other, smaller issues:
 - Eq (4) should be written as a matrix inverse, not a fraction.
 - In the UKF the Julier paper of 1997 also notes a heuristic solution for ensuring positive definiteness of the estimated covarance matrix is lambda is negative.  Was this tried?
 - How was the number of particles selected for the PF at test time?  In particular, how did the computation time between the methods compare?

---

> ### Author Response · Authors · 2018-11-24
> **Response**
>
> Thanks a lot for the helpful review. We agree that the suggested baseline experiment would be very useful and will try to add it in future versions. We can unfortunately not include it within the rebuttal period.
>
> Replies to your questions:
>
> * In the UKF the Julier paper of 1997 also notes a heuristic solution for ensuring positive
> definiteness of the estimated covariance matrix is lambda is negative. Was this tried?
>
> Reply:
> No, we did not try this. However, another main problem that we found with negative lambda (while lambda + n is positive) is that the sigma point for the mean is given negative weight while the other points receive positive weights. This can make the mean estimate diverge on its own and also does not make a lot of sense intuitively. We added this in the updated version.
> In general we would therefore recommend using positive lambda to avoid both problems.
>
>
> * How was the number of particles selected for the PF at test time?  In particular, how did the computation time between the methods compare?
>
> Reply:
> We chose the number of test particles to be the same as was used in (Jonschkowski et al., 2018) to ensure comparability on the kitti task.
> Although we did not attempt timing experiments, the test-time did not seem to increase much
> between 100 and 1000 particles, as long as the computations for each particle can still be run in parallel. This is of course dependent on the GPU in use. Due to this parallelism, there also seems to be no big difference between the computation times for the different filtering methods.
>
>
> References:
>
> Rico Jonschkowski, Divyam Rastogi, and Oliver Brock. Differentiable particle filters: End-to-end learning with algorithmic priors. In Proceedings of Robotics: Science and Systems, Pittsburgh, USA, 2018.

---

### Official Review · AnonReviewer2 · 2018-11-04
**Nice study which (sadly) ignores large parts of the related work**

**Rating:** 6
**Confidence:** 5

**Review:**

# Review for "On Learning Heteroscedastic Noise Models within Differentiable Bayes Filters"

The method revisits Bayes filters. It evaluates the benefit of training the observation and process noise models, while keeping all other models fixed. Experimentally, a clear performance boost is verified if heteroscedastic noise is used.

First, I want to applaud the effort done to do the study. I think it is very beneficial for the community to revisit classic algorithms and evaluate them in a broader and more recent context. I certainly will revisit this article and point colleagues to it.

The paper is well-written and the experiments seems to be well done. The review of the relevant models is adequate, although space filling, since the methodology  is not at the core of ICLR. I however consider it highly relevant for the future of the field.

However, there is a major flaw: the variational state-space model literature is completely ignored. I consider this blank spot unacceptable. Especially, the models proposed have already explored heteroskedastic noise models in contexts where state-space models and posterior approximations were fully trained. It is just that an ablation study was never done.

I am very torn, as I like the paper in general but think that the recognition of the variational SSM literature needs to be added, and not having it in here would foster a separation of two "micro communities".

Here is an incomplete list of articles, which can serve as starting points for a more thorough literature review.

- Archer, E., Park, I. M., Buesing, L., Cunningham, J., & Paninski, L.
(2015). Black box variational inference for state space models. arXiv preprint arXiv:1511.07367.
- Fraccaro, M., Sønderby, S. K., Paquet, U., & Winther, O. (2016). Sequential neural models with stochastic layers. In Advances in neural information processing systems (pp. 2199-2207).
- Karl, M., Soelch, M., Bayer, J., & van der Smagt, P. (2016). Deep  variational bayes filters: Unsupervised learning of state space models from rawdata. ICLR 2017.

---

> ### Author Response · Authors · 2018-11-24
> **Response**
>
> Thanks a lot for the positive and encouraging feedback on our work! We highly appreciate the pointer
> to the variational bayes methods and agree that not mentioning them was an oversight on our part. We have updated the related works section accordingly.
> In future work, it would be very interesting to see how our approach compares to a variational method. Running such experiments is however not possible during the rebuttal period.

---

### Author Response · Authors · 2018-11-27
**Comment on the revised version**

We want to again thank our reviewers for their helpful and kind feedback. This comment provides a short summary of our contributions and the changes we made in the revised version of the paper.

Contributions:
In this work, we analyzed the advantages of learning heteroscedastic models of observation and especially process noise through differentiable bayesian filters.
For this, we evaluated training the noise models through 4 different filtering algorithms: Differentiable versions of Extended Kalman Filter and Particle Filter had already been proposed in related work (Haarnoja et al., 2016, Jonschkowski et al., 2018, Karkus et al., 2018), but we also added  differentiable versions of two Unscented Kalman Filter variants. In addition to comparing within the different filters, we also evaluated learning the noise models on two different tasks: The Kitti Visual Odometry task has a low-dimensional state and smooth dynamics. In contrast, the planar pushing task has discontinuous dynamics and is known to follow a heteroscedastic noise model. It also has a much higher-dimensional state than any other task evaluated in related work.

Summary of revisions:
- Minor changes following the questions of our reviewers to clarify things or add information
- Following the suggestion of Reviewer 1, we added a section about variational methods for learning in state space models to the related works section. They offer an alternative approach to learning as compared to our method of backpropagation through differentiable filtering algorithms. Experiments that compare both approaches would be very interesting, but could not be carried out within the rebuttal period.
- We updated the results for the Kitti experiments to use the full testset available. To improve the comparability of the obtained results with related work (Haarnoja et al., 2016, Jonschkowski et al., 2018), we changed the experimental setting to allow for finetuning of the perception network during training. This improved the overall results on the full testset, but did not change the conclusions drawn from the experiment. We also rewrote the discussion of the results to better explain the differences in performance between the different filtering algorithms.

References:
Tuomas Haarnoja, Anurag Ajay, Sergey Levine, and Pieter Abbeel. Backprop kf: Learning discriminative deterministic state estimators. In Advances in Neural Information Processing Systems. 2016

Rico Jonschkowski, Divyam Rastogi, and Oliver Brock. Differentiable particle filters: End-to-end learning with algorithmic priors. In Proceedings of Robotics: Science and Systems, Pittsburgh, USA, 2018.

Peter Karkus, David Hsu, and Wee Sun Lee. Particle filter networks: End-to-end probabilistic localization from visual observations. 2018

---

### Meta-Review · Area_Chair1 · 2018-12-16
**Interesting but not good enough.**

**Confidence:** 5
**Recommendation:** Reject

**Metareview:**

This paper shows experiments in favor of learning and using heteroscedastic noise models for differentiable Bayes filter. Reviewers agree that this is interesting and also very useful for the community. However, they have also found plenty of issues with the presentation, execution and evaluations shown in the paper. Post rebuttal, one of the reviewer increased their score, but the other has reduced the score. Overall, the reviewers are in agreement that more work is required before this work can be accepted.

Some of existing work on variational inference has not been included which, I agree, is problematic. Simple methods have been compared but then why these methods were chosen and not the other ones, is not completely clear. The paper definitely can improve on this aspect, clearly discussing relationships to many existing methods and then picking important methods to clearly bring some useful insights about learning heteroscedastic noise. Such insights are currently missing in the paper.

Reviewers have given many useful feedback in their review, and I believe this can be helpful for the authors to improve their work. In its current form, the paper is not ready to be accepted and I recommend rejection. I encourage the authors to resubmit this work.